# The Fecal Redox Potential in Healthy and Diarrheal Pigs and Their Correlation with Microbiota

**DOI:** 10.3390/antiox13010096

**Published:** 2024-01-12

**Authors:** Ni Feng, Rongying Xu, Dongfang Wang, Lian Li, Yong Su, Xiaobo Feng

**Affiliations:** 1Laboratory of Gastrointestinal Microbiology, Jiangsu Key Laboratory of Gastrointestinal Nutrition and Animal Health, College of Animal Science and Technology, Nanjing Agricultural University, Nanjing 210095, China; 2022205037@stu.njau.edu.cn (N.F.); 2020205027@stu.njau.edu.cn (R.X.); 2023205044@stu.njau.edu.cn (D.W.); lilian@njau.edu.cn (L.L.); 2Research Institute of General Surgery, Jinling Hospital, Nanjing University School of Medicine, Nanjing 210002, China

**Keywords:** redox potential, weaning diarrhea, intestinal microbiota, correlation analysis

## Abstract

The redox potential plays a critical role in sustaining the stability of gut microbiota. This study measured the fecal redox potential in healthy and diarrheal pigs using direct and dilution methods and investigated their correlation with microbiota. The results showed that the fluctuations in the redox potential of healthy pig feces were consistent using two different methods and the two methods are equivalent based on an equivalence test. The redox potential was positively correlated with the number of fungi and negatively related to the total bacteria. The relative or absolute abundances of many bacteria at the phyla and genus levels were associated with redox potential. In diarrheal pigs, the potentiometric trends of the two methods demonstrated an opposing pattern and the correlation with total bacteria was reversed. Precipitously elevated redox potential was detected post-diarrhea using dilution methods. The absolute abundance of *Escherichia-Shigella* and *Fuurnierella* was positively correlated with redox potential, while both relative and absolute abundances of *Limosilactobacillus* were positively correlated. These results suggest that both methods are suitable for detecting gut redox potential in healthy pigs, while the dilution method is more suitable for diarrheal pigs. The findings on the correlation of *Limosilactobacillus*, *Prevotella,* and *Escherichia-Shigella* with redox potential offer novel insights for targeted modulation of intestinal health.

## 1. Introduction

The composition of the intestinal microbiota is influenced by various factors within the gut environment, including substrates, growth factors, micronutrient availability, concentrations of antimicrobial compounds, redox potential, and intestinal pH levels [1,2]. The redox potential serves as a macroscopic indicator of the complex reactions occurring within the gut environment and it is intimately associated with microbial reactions and intestinal health. Significant alterations in the dietary composition of animals occur during different growth stages, resulting in corresponding modifications to the intestinal microbiota and the intestinal environment [3]. However, the temporal changes in gut redox potential with the growth of an animal have not been reported to date.

The gut environment Is characterized by a complex mixture of bacteria, chyme, and various metabolites. These bacteria actively modify their chemical environment within the gut, allowing pathogenic bacteria to proliferate under high redox potential conditions and recover more rapidly following antibiotic treatment [4]. Although the chemical environment of the gut was restored following antibiotic treatment, the eradication of certain bacterial species suggests that the chemical milieu is dictated by the bacterial communities inhabiting the gut. Redox potential, which captures numerous aspects of an environment’s chemical composition, provides an estimation of the growth efficiency of specific bacteria within that environment [5]. The measurement value of intestinal redox is influenced by various measurement methods. Currently, the measurement of intestinal redox potential can be categorized into two methods. One method involves directly measuring the intestinal redox potential during anaerobic operations (direct method) [6]. The measurement technique is unaffected by oxygen presence and accurately depicts the actual redox status of the intestine. Nonetheless, it is notably influenced by the water of intestinal contents. In cases of diarrhea, where there is an elevated water content in the intestines, the obtained value may contradict the actual intestinal redox state. The other method involves freeze-drying the intestinal contents to achieve a standardized water content (dilution method) [7]. The measured value of this method may be susceptible to oxygen interference, resulting in an overestimation of the intestinal redox potential. Consequently, it is imperative to investigate which measurement approach can provide a more accurate reflection of the intestinal redox state under various conditions.

Postnatal microbial colonization of the animal gastrointestinal tract progressively establishes and stabilizes over time. Alterations in the intestinal microbial composition can impact the health and growth performance of pigs. Gryaznova et al.’s [8] metagenomic analysis of the fecal microbiota in piglets with diarrhea demonstrated the presence of *Prevotella*, *Surella*, *Campylobacter*, and Fusobacteriaceae bacteria in the feces of diarrheal piglets, suggesting a significant association between these microbial alterations and the occurrence of diarrhea. To obtain a more comprehensive understanding of the abundance of specific taxa, an alternative approach involves conducting real-time quantitative PCR with custom-designed primers targeting various bacteria. In the context of differential abundance analysis, the presence of component effects in the relative abundance data may give rise to spurious differences (false positives) when assessed in absolute terms. Absolute and relative data consistently reveal traits that exhibit substantial and well-defined modifications under diverse conditions [9]. Real-time quantitative PCR serves as a precise technique for quantitatively determining absolute gene abundance and thus, integrating real-time quantitative PCR with high-throughput sequencing enables more robust characterization of the actual shifts occurring within microbial populations [10]. When diarrhea occurs, there is a significant increase in the intestinal redox potential [11]. The increase in redox potential leads to a state of oxidative stress in the intestinal tract which, in turn, results in intestinal environmental disorders and exacerbates the damage to the intestinal health, thus adversely affecting the intestinal environment. The intestinal microbiota undergo changes during episodes of diarrhea [12], indicating fluctuations in both redox potential and intestinal microbiota in diarrheal individuals. However, the relationship between redox potential and intestinal microbiota remains unclear. Therefore, this study aimed to compare fecal redox potential in healthy and diarrheal pigs by using direct or dilution methods and investigate the correlation of redox potential with specific gut microbiota based on the relative or absolute abundance. The study will offer novel ideas and understanding for regulating gut microbiota and health of animals.

## 2. Materials and Methods

### 2.1. Animals, Experimental Design and Sampling

The animal study protocol was approved by the Animal Care and Use Committee of Nanjing Agricultural University (SYXK2019-0066) and implemented based on the standard of Experimental Animal Care and Use Guidelines of China (EACUGC2018-01). The experiment was conducted in Zhenjiang City, Jiangsu Province. Ten litters of piglets with similar birth dates were selected. The piglets were housed together with the sows from birth until weaning and then transferred to individual pens in the nursery after 28 days of weaning. None of the pigs received antibiotics or probiotic products throughout the study period. After weaning, commercial feed consisting mainly of corn, soybean meal, soybeans, and soybean oil (Appendix A) was provided for the conservation pigs. The feed formula standard during the growing and finishing periods is shown in Appendix A. All pigs had libitum access to feed and water. Over the same period, three pigs with similar body weights were randomly selected from each litter (or pen) at 10 days (lactation period), 40 days (conservation period), 100 days (growth period), and 150 days (fattening period) to collect fresh fecal samples. Samples collected on the same day from the same litter (or pen) were pooled together; ten litters or pens were collected for each growth phase. Piglets exhibiting spontaneous post-weaning diarrhea and unaffected counterparts were carefully chosen from distinct litters within a single swine facility. Fresh fecal samples were collected in a CO_2_-sealed bag and the sampling time for each piglet was strictly controlled within a half-hour interval. Immediately after collection, the samples were securely sealed and thoroughly mixed, followed by the determination of redox potential. Samples for microbiota analysis were collected using RNase- and DNase-free freeze-storage tubes and stored initially in liquid nitrogen before being transferred to a refrigerator at −80 °C.

### 2.2. Redox Potential Measurement

Direct method: According to study [6], CO_2_ was injected into the sealed bag. Fresh feces was collected and placed in the same bag, followed by the insertion of the potentiometer’s electrode directly into the test specimen. ST300/B ORP electrodes were employed to assess fresh fecal samples, with all measurements conducted under anaerobic conditions.

Dilution method: The redox potential was determined utilizing a modified experimental protocol with ST300/B ORP electrodes [7]. Initially, the frozen fresh feces were diluted in distilled water at a 1:10 ratio and then agitated via vortex. Subsequently, the mixture was centrifuged at 8000 rpm for 10 min, followed by the measurement of redox potential in a random order. The electrode does not need to be calibrated because of its built-in standard electrode. After each sample was measured, the electrode was repeatedly rinsed with deionized water, and the next sample was measured after a light wipe with a paper towel.

### 2.3. DNA Extraction and Real-Time Quantitative PCR

According to previous research [13], a comprehensive genomic DNA extraction protocol was employed, which involved isolating DNA from feces in conjunction with the guidelines provided by a fecal DNA extraction kit. The concentration of the extracted DNA was determined using a nanodrop spectrophotometer (Thermo, Wilmington, DE, USA). Real-time quantitative PCR was performed with specific primers of total bacteria [14], fungi [15], and methanogens [16] by an ABI 7300 sequence detector (SDS, Foster City, CA, USA) using the SYBR Green PCR kit (TaKaRa, Co., Dalian, China). Employing the method suggested by Jing et al. [14], a panel of 16S rRNA genes representing bacterial monoclonal genes was selected for plasmid preparation. After establishing a standard curve through gradient dilution, the total bacterial content in the sample was calculated using the standard curve. The fungal-specific primers used for real-time PCR assay were designed from multiple alignment of ITS-1 gene sequences, which are representatives of distantly related fungal species [15]. Amplification efficiencies for the methanogens real-time quantitative PCR assay were performed on plasmid DNA purified from clone CLI06 and CLI09 from the clone library generated from the rumen as described above [17]. The results are expressed as the numbers of 16S rRNA or ITS gene copies per gram of wet feces.

### 2.4. 16S rRNA Gene Sequencing and Data Processing

The V4-V5 regions of the bacteria 16S ribosomal RNA gene were amplified by PCR (95 °C for 2 min, followed by 25 cycles at 95 °C for 30 s, 55 °C for 30 s, and 72 °C for 30 s and a final extension at 72 °C for 5 min) using primers 515F 5′-barcode- GTGCCAGCMGCCGCGG)-3′ and 907R 5′-CCGTCAATTCMTTTRAGTTT-3′, where the barcode is an eight-base sequence unique to each sample. PCR reactions were performed in triplicate with 20 μL mixture containing 4 μL of 5 × FastPfu Buffer, 2 μL of 2.5 mM dNTPs, 0.8 μL of each primer (5 μM), 0.4 μL of FastPfu Polymerase, and 10 ng of template DNA. Amplicons were extracted from 2% agarose gels and purified using the AxyPrep DNA Gel Extraction Kit (Axygen Biosciences, Union City, CA, USA) according to the manufacturer’s instructions. Amplicon Sequence Variants (ASVs) were clustered utilizing UPARSE version 7.1, with a similarity cutoff of 97% [18]. The operational taxonomy was hierarchically clustered using UPARSE and the ribosome database item classifier was leveraged for systematic affinity analysis of each representative ASV sequence. The sparse curves of 16S rRNA gene sequences exhibit a tendency towards saturation platforms, ensuring sufficient sequencing depth for diversity analysis. Bacterial diversity assessment encompassed Chao1 and observed species. Each representative ASV sequence was rigorously examined by the ribosome database item classifier and assigned an confidence level of 80% [19]. Principal coordinate analysis and non-metric multidimensional scaling analysis were performed utilizing the Bray–Curtis distance as a basis. The biomarkers within the microbiome data were determined employing the online Wilcoxon analytical diarrhea group method. Functional prediction analysis of gene fragments was performed using FAPROTAX.

### 2.5. Statistical Analysis

The data statistics were generated by SPSS22.1 and a one-way ANOVA was employed for the comparison of fecal redox potential of ten pigs at the ages of 10, 40, 100, and 150 days, while a *t*-test was utilized to examine the differences between post-weaning (40 d) healthy and natural diarrhea after weaning piglets. Litter or pen was regarded as one experimental unit (*n* = 10) of all analyses. The equivalence test of direct and dilution redox potential measurement methods was carried out with Minitab 23. The outcome of the tests was represented as “mean ± standard error”, and the criterion for determining significant differences was set at *p* < 0.05. The absolute abundance of various bacterial genera was calculated by multiplying the relative abundance of the bacterial community by the total number of bacteria. The correlation between specific bacteria and redox potential was analyzed using Pearson correlation based on the relative or absolute abundance.

## 3. Results

### 3.1. The Correlation of Fecal Redox Potential with the Number of Bacteria, Methanogens, and Fungi in Healthy Pigs

The fecal redox potential fluctuated with age coincidently by using direct and dilution methods (Figure 1a,b), and the redox potential of 40-day-old pigs was significantly increased by both direct and dilution methods compared with 10-day-old pigs (*p* < 0.05). Oppositely, the redox potential demonstrated a significant reduction in 100-day-old pigs compared with 40-day-old pigs (*p* < 0.05). In contrast, an increasing trend was observed for the redox potential in 150-day-old pigs compared with 100-day-old pigs (*p* < 0.05). Pearson correlation analysis revealed a significant positive correlation between the two measurements of redox potential (r = 0.7723; *p* < 0.001; Figure 1c). A 90% CI for mean (dilution method)–mean (direct method) CI was within the equivalence interval (Figure 1d). The numbers of intestinal microbiota across different growth stages demonstrated variations in total bacterial, with significantly lower abundance observed at day 150 compared to other stages (*p* < 0.05; Figure 1e). However, total fungi was notably elevated in piglets at day 40 compared to other growth stages (*p* < 0.05; Figure 1f). Furthermore, methanogens demonstrated a significant increase in piglets at day 40 in comparison to those at day 100 (*p* < 0.05; Figure 1g). Pearson correlation analysis revealed a negative correlation between the redox potential and total bacterial as measured by both two methods (Figure 1h,k), with a stronger association observed for the dilution method (r = −0.5031; *p* = 0.0009; Figure 1k). The measured values for redox potential using both two methods exhibited a positive correlation with fungi (Figure 1i,l); however, the direct method demonstrated a stronger association (r = 0.5287; *p* = 0.0005; Figure 1i). The redox potential also displayed a positive correlation with methanogen (r= 0.4507; *p* = 0.0035; Figure 1j) using the direct method, with no significant correlations using the dilution method (Figure 1m).

### 3.2. Bacterial Communities in Feces of Pigs at Different Growth Stages

The Chao1 index and observed species of fecal microbiota progressively increased with age (*p* < 0.05; Figure 2a,b) and a significant differentiation in bacterial communities was observed at different ages (*p* < 0.05; Figure 2c,d). Furthermore, the succession of intestinal microbiota was found to change significantly with the advancement of pig age (Figure 2e,f). At the phylum level, Firmicutes emerged as the dominant phylum across growth stages. The proportion of Actinobacteriota and Spirochaetota in piglets weaned at 40 days increased significantly compared with those at 10 days (*p* < 0.05; Figure 3a). The relative abundance of *Limosilactobacillus* exhibited a significant decline (*p* < 0.05; Figure 3b) except for 10 days. The application of FAPROTAX enabled the prediction of microbial function, indicating a significant alteration in the function of fecal microbiota before and after weaning, as well as a notable increase in chemoheterotrophy and fermentation functions at different growth stages (*p* < 0.05; Figure 3c). Notably, the microbial function of 40-day pigs with low redox potential demonstrated a significant decline in nitrite ammonification, nitrite respiration, and nitrogen respiration (*p* < 0.05; Figure 3d) compared to that of 10-day pigs. Furthermore, this function exhibited a consistently lower trend compared to other growth stages.

### 3.3. Correlation of Fecal Redox Potential with the Abundance of Bacterial Community at Phylum and Genus Levels

The correlation analysis revealed significant associations between the redox potential and various species in terms of both relative and absolute abundance. Specifically, nine phyla demonstrated significant positive correlations with the redox potential in their absolute abundance, while eight species exhibited a significant positive correlation in their relative abundance. A Venn diagram further elucidated a significant correlation among five specific species and the redox potential (Figure 4a). Furthermore, Spirochaetota, Fibrobacterota, and Cyanobacteria displayed positive correlations with redox potential (Figure 4c,d), although this association gradually attenuated when considering both their relative and absolute abundances. Conversely, a negative correlation was observed between Verrucomicrobiota, Synergistota, and the redox potential (Figure 4e,f). Remarkably, the relative abundance of Verrucomicrobiota exhibited a more pronounced negative correlation with the redox potential (r = −0.3877; *p* = 0.0134; Figure 4f), whereas the negative correlation between the absolute abundance of Synergistota and the redox potential was even stronger (r = −0.4357; *p* = 0.0049; Figure 4f). At the genus level, the redox potential was observed to have significant correlations with the absolute abundances of 103 bacterial genera and the relative abundances of 70 bacterial genera. Additionally, 44 bacterial genera were found to be associated with redox potential both in the absolute and relative abundances (Figure 4b). The top five bacterial genera positively associated with redox potential included *Catenisphaera*, *Enterorhabdus*, *paludicola*, *papillibacter*, and *UCG-009* (Figure 5a,b). In contrast, the top five bacterial genera exhibiting a negative correlation were *Sellimonas*, *Anaertruncus*, *Veillonella*, *Intestinimonas*, and *Odoribacter* (Figure 5c,d). Specifically, *Catenisphaera* demonstrated the strongest positive correlations with redox potential for both its relative and absolute abundance. The relative abundance of *Anaertruncus* displayed the highest negative correlation with redox potential, whereas the absolute abundance of *Anaertruncus* exhibited the second highest negative correlation with redox potential.

### 3.4. Fecal Redox Potential, the Numbers of Bacteria, Methanogens, and Fungi in Diarrheal Piglets

The determination of redox potential values using both two methods exhibited a positive correlation across varying growth stages. Upon the occurrence of diarrhea, modifications in stool morphology and an increase in water content were observed. The direct method directly revealed a significant influence of stool water content on the redox potential, thereby rendering it unable to directly reflect the intestinal redox status. To circumvent discrepancies induced by elevated fecal water content during diarrhea (Figure 6a), the dilution method and literature approaches were implemented to determine the redox potential value (Figure 6b). In instances of diarrhea, a minimal correlation was observed between the two methods (r = −0.3033; *p* = 0.1936; Figure 6c), but the equivalence test of direct and dilution methods was equivalent. Post-diarrhea, a significant escalation in the total bacterial was noted (*p* < 0.05; Figure 6e), concurrently accompanied by a significant decline in methanogens (*p* < 0.05; Figure 6g). No significant alteration was detected in the fungi (*p* > 0.05; Figure 6f). The subsequent correlation analysis revealed that the redox potential derived from the dilution method was positively correlated with the total number of bacteria (r = 0.3247; *p* = 0.0457; Figure 6k), and negatively correlated with methanogens (r = −0.5550; *p* = 0.0111; Figure 6m). However, no correlation was observed using the direct method (Figure 6j). No association was detected between fungi and redox potential measurements for the direct and dilution methods (Figure 6i,l).

### 3.5. Correlation of Fecal Redox Potential with the Abundance of Bacterial Community in Diarrheal Piglets

Although the total number of bacteria increased in the diarrhea group, a significant decrease in the diversity of bacterial community was observed (*p* < 0.05; Figure 7a,b); however, the intestinal microbiota could not be completely separated from the normal group (Figure 7c). In the diarrhea group, a significant increase was observed in the relative abundance of Firmicutes, accompanied by a significant decrease in the relative abundance of Bacteroides (Figure 7d). Furthermore, Firmicutes (r = 0.5919; *p* = 0.0060) exhibited a positive correlation with redox potential, whereas Bacteroidetes (r = −0.6728; *p* = 0.0012; Figure 7e) demonstrated a negative correlation with redox potential. At the phylum level, except for Firmicutes and Bacteroides, the absolute abundance of Spirochaetota (r = −0.4532; *p* = 0.0448) displayed a negative correlation with the redox potential, whereas the absolute abundance of Proteobacteria (r = 0.4854; *p* = 0.0300; Figure 8a,b) exhibited a positive correlation with the redox potential. The differential microflora between the diarrhea group and the control group encompass *Dorea*, *UCG-005*, *UCG-002*, *Oscillospira*, *Roseburia*, *Lachnospira*, *Rikenellaceae RC9 intestinal group*, *parabacteroides*, and *Clostridium sensu stricto 6* (Figure 7f). *Limosilactobacillus* (r = 0.5811, *p* = 0.0072, Figure 7g), *Dorea* (r = 0.5570, *p* = 0.0107, Figure 7g), and *Lachnospira* (r = 0.4617, *p* = 0.0404, Figure 7g) exhibited a positive correlation with the redox potential. The absolute abundance of *Limosilactobacillus* (r = 0.4530; *p* = 0.0449; Figure 8d) was also positively correlated with the redox potential. Notably, a positive correlation was observed between *Escherichia-Shigella* (r = 0.5304; *p* = 0.0161; Figure 8d) and the intestinal redox potential. In the event of diarrhea, the numbers of intestinal aerobic pathogens escalated in concurrence with the increase of redox potential.

### 3.6. The Association between Redox Potential and the Fecal Microbial Community of Pigs with Diarrhea at the ASV Level

In addition, we further analyzed the association between redox potential and fecal microbial communities of pigs with diarrhea at the ASV level. Correlation analysis identified 111 types of ASVs that were significantly correlated with redox potential, with 73% showing positive correlation and 27% showing negative correlation (Figure 9a). In addition, bubble charts were utilized to visually present the top ten ASVs that exhibited a positive or negative correlation with redox potential in the diarrhea and control group (Figure 9b). Notably, *Limosilactobacillus* ASV 20 demonstrated a significant increase in abundance within the diarrhea group (*p* < 0.05; Figure 9c) and was positively associated with redox potential (r = 0.5291; *p* = 0.0164; Figure 9c). ASV represents the species level and each ASV represents a bacterium, which may only be identified as a genus. The mulberry map shows the species of bacteria annotated by ASV (Figure 9d). The findings revealed that the *Limosilactobacillus* strain exhibited consistent variations compared to ASV20, which were also supported by Wilcoxon analysis and positively correlated with redox potential (Figure 9e). Meanwhile, the annotation of *Prevotella 7* ASV 60 (r = −0.5834; *p* = 0.0069) and *Prevotella 9* ASV 5, 26, 67 (r = −0.5733; *p* = 0.0082) was negatively associated with redox potential.

## 4. Discussion

Numerous intricate internal and external factors exert influences on the succession and structure of intestinal microorganisms [20]. The equilibrium of bacterial species and their metabolites in the gut significantly influences the preservation of intestinal health or the progression of diseases [21]. The impacts of various factors within the intestinal environment on the homeostasis of intestinal microbiota, including redox potential is a subject of ongoing research [5]. The redox potential represents a macroscopic manifestation of complex reactions within the intestinal environment, and its fluctuations are intimately linked to intricate bacterial reactions and the overall health of the intestinal system [22]. This study uncovered the fluctuations in intestinal redox potential throughout various growth stages, which potentially reflected the succession of intestinal flora during these stages. Moreover, within the same growth stage, the intestinal redox potential also changed with and without diarrhea and the intestinal flora also fluctuated. Weaning stress often causes changes in the morphology and function of the small intestine of piglets, disrupts digestion and absorption capacity, destroys intestinal barrier function, and ultimately leads to reduced feed intake, increased diarrhea rate, and growth retardation [23]. Diarrhea is a prevalent manifestation of weaning stress and can also induce alterations in the intestinal microbiota composition of piglets. Yang et al. [24] demonstrated that diarrheal piglets exhibited lower relative abundances of *Bacteroides*, *Ruminococcus*, *bullobacter*, and *Treponemas* compared to healthy piglets post-weaning, with these genera playing a pivotal role in nutrient metabolism. Diarrhea frequently occurs concurrently with inflammation, thus resulting in modifications to the intestinal microbiota’s composition by promoting bacteria that thrive in inflammatory and aerobic environments. In this disharmonious setting, electron receptors can be supplied for metabolic processes within the body, thus prompting microorganisms to favor metabolic reactions with elevated thermodynamic energy derived from redox potential [25]. The redox potential in diarrhea experienced a significant increase. Conversely, anaerobic environments promoted metabolic reactions that possessed lower oxidation–reduction potential energy in healthy organisms, thermodynamically speaking [26]. Redox potential was effectively regulated at a minimal level. Intriguingly, gut microbiota exhibited corresponding fluctuations, and redox potential was a macroscopic value that mirrored the redox state of the intestinal environment. The abundance of Bacteroidota in the diarrhea group was significantly reduced, consistent with previous studies. Correlation analysis revealed a negative association between redox potential and Bacteroidota, providing an explanation for the elevated redox potential observed in the diarrhea group (Figure 7). Enzyme activity, cellular assimilation capacity, and microbial growth and development were also influenced by redox potential. The diversity of intestinal microbiota in piglets significantly increased post-weaning (40 d) and reached a higher level as pigs grew and developed (Figure 2). It is well-documented that an increased diversity of gut microbiota enhances the stability and functionality of intestinal flora. Notably, it has been suggested that gut microbiota diversity is a novel biomarker for health and metabolic capacity [27]. The occurrence of diarrhea significantly reduced the diversity of intestinal flora (Figure 7), which indicated a substantial decline in the stability and function of the intestinal tract. After pig birth, upon exposure to a variety of bacteria in the environment, the number and diversity of intestinal flora rapidly increased, which was crucial for the development of intestinal flora. The outcomes of the non-metric multidimensional scaling analysis demonstrated that the composition of the intestinal microbiota varied, which suggested that the structure and composition of the intestinal microbiota might undergo significant alterations over time (Figure 2). In essence, bacteria, fungi, and methanogens represent the primary microbial kingdoms engaged in the fermentation of dietary fibre within the intestine [6]. The bacteria in the gut include beneficial bacteria, harmful bacteria, and conditioned bacteria. Their activity and mediator molecules contribute to our health [28]. Most fungi thrive in aerobic conditions. However, within the anaerobic environment of the intestinal tract, fungal populations are typically low. The dysregulation of intestinal fungal composition has been implicated in various gastrointestinal disorders [29]. The majority of gut-associated archaea exhibit a distinctive metabolic process wherein they utilize the byproducts of bacterial fermentation, including hydrogen, carbon dioxide, ormats, acetate, and others. Through this metabolic pathway, methanogenic archaea enhance the efficiency of bacterial metabolism and thereby significantly facilitate energy transfer among microorganisms within the host organism [30]. The quantitative findings demonstrated significant disparities in the total number of bacteria, fungi, and methanogens across various growth stages (Figure 1). In the event of diarrhea, the total bacteria surged, whereas the fungi remained relatively unchanged, and the methanogens observably decreased. In conjunction with the analysis of microbiota structure and diversity, the study revealed an increase in the number of intestinal flora, while a decline in flora diversity during diarrhea (Figure 6).

The results of the correlation analysis demonstrated a negative correlation between the total number of bacteria in various growth stages and the redox potential, whereas a positive correlation was observed between the total number of methanogens and the redox potential (Figure 4). Additionally, a positive correlation between the total number of bacteria and the redox potential was noted during diarrheal episodes (Figure 8), which suggests that the intestinal microbiota composition varies between healthy and pathological states. The relative and absolute abundances of Spirochaetota, Fibrobacterota, Cyanobacteria, Verrucomicrobiota, and Synergistota were found to be correlated with redox potential at varying growth stages. Notably, Fibrobacterota was involved in the production of short- and medium-chain fatty acids (acetic acid, butyric acid, and propionic acid). Surprisingly, Fibrobacterota exhibited a robust metabolic capacity, including the synthesis of acetic acid, alanine, histidine, arginine, tryptophan, serine, threonine, valerian, and other amino acids. These metabolites contributed to the host’s resistance against hypoxia [31]. The Verrucomicrobiota primarily inhabited the luminal layer of the intestinal mucosa, which corresponded to the sole species of *Ackermannia* currently recognized [32]. This microbial species was prevalent in healthy individuals, with the capacity to decompose polysaccharide substances, such as mucosaccharides and cellulose, aiming to provide energy and nutrients. Obesity, inflammatory bowel disease, and type 2 diabetes, among other conditions, were associated with a reduced abundance of Verrucobacteria [33], which indicated that the increase of redox potential can reduce the abundance of anaerobic beneficial bacteria in the intestinal tract. The reduction of redox potential can enhance fermentation metabolism of customized tactics that defend these various enzymes from occasional oxygen exposure [34]. The relative and absolute abundances of *Catenisphaera*, *Enterorhabdus*, *Paludicola*, *Papillibacter*, and *UCG-009* were positively correlated with redox potential (Figure 5), which suggests that a moderate increase in redox potential within a healthy collective is also beneficial for the oxidative metabolic reactions of beneficial bacteria. Due to their negative correlation with redox potential, the microbial genuses *Sellimonas*, *Anaertruncus*, *Veillonella*, *Intestinimonas*, and *Odoribacter* warrant special attention. Notably, the abundance of *Sellimonas* was significantly augmented in patients with anxiety disorders [35]. The redox potential of the organism corresponded to a decrease in the abundance of *Anaerotruncus*. *Anaerotruncus* utilize lactic acid as its exclusive carbon source to enhance the abundance of marathon runners post-exercise, thereby boosting the endurance of athletes [36]. Similarly, *Intestinimonas* and *Odoribacter* also contribute to anaerobic intestinal metabolism [37].

Empirical research has consistently demonstrated that a disruption in the intestinal microbiome equilibrium significantly contributes to diarrheal pathogenesis [38]. The preservation of the intestinal microbiome diversity and stability is crucial for sustaining the homeostasis of the intestinal microecosystem. In recent years, the study of gut microbiota has garnered the attention of researchers worldwide, and it has been established that gut microbiota plays a role in and impacts the physiological and pathological processes of the human body. The primary functions of gut microbiota encompass carbohydrate metabolism, cellulose degradation, immune response, and the regulation of neurophysiological functions [39]. The imbalance of intestinal microbiota was primarily attributed to the decline in probiotic levels and horizontal transfer of bacterial flora distribution. Decreased diversity of intestinal flora, shifts in composition, increased consumption of carbohydrates, or changes in dietary structure induced by various factors can result in abnormal carbohydrate metabolism. This, in turn, leads to an increase in osmotic pressure of intestinal fluid, disrupted absorption of water, and ultimately nutritional diarrhea [40,41]. When diarrhea occurred after weaning, correlation analysis showed that the relative and absolute abundance of *Limmosilactobacillus* in different strains were positively correlated with redox potential (Figure 8). *Limmosilactobacillus* has the potential to ameliorate inflammatory markers and oxidative stress indicators associated with hyperuricemia, thereby serving as a supplementary agent in the management of hyperuricemia [42]. The detrimental consequences of intestinal oxidative stress were intricately linked to the impairment of intestinal microbiota and the disruption of intestinal ecological homeostasis. *Limosilactobacillus* has demonstrated the presence of a potent antioxidant mechanism, which offers protection against oxidative stress and its associated chronic ailments [43]. In the meantime, empirical research has demonstrated that *Limosilactobacillus* can effectively mitigate the symptoms of diarrhea and colon inflammation induced by antibiotics, concurrently facilitating the normal expression of colon immune factors [37], which shows that the gut flora can cascade with the body’s immune system to fight the disease when the body has diarrhea. It was found that the abundance of *Limosilactobacillus reuteri* ASV increased in the diarrhea group (Figure 9). As one of the primary metabolites produced by *Limosilactobacillus reuteri*, Reuterin exhibits inhibitory effects on harmful bacteria, such as *Escherichia-Shigella* and *Salmonella*, thereby safeguarding the intestinal microecological balance. In this study, a positive correlation was observed between the redox potential detected in cases of diarrhea and the abundance of *Limosilactobacillus* and *Escherichia-Shigella*, which indicated a harmonious equilibrium within the intestinal flora. In the event of diarrhea, the absolute abundance of *Blautia* demonstrated a negative correlation with the redox potential. It has been shown that *Blautia*, one of the most prevalent and crucial acetogenic bacteria in the gut, could alleviate depression and accelerate the progression of breast cancer [44]. Consequently, an increase in gut redox potential leads to a decline in the abundance of key probiotics in the gut in cases of diarrhea. The jury is still deliberating the potential benefits or drawbacks of *Prevotella*. *Prevotella* has demonstrated positive effects on glucose regulation and host metabolism [45]. Conversely, *Prevotella* is associated with local or systemic infections [46]. The negative correlation observed between *Prevotella* and redox potential in this study can likely be attributed to the increased diversity of *Prevotella*, which enhances the fermentation capacity of the microbiome and promotes intestinal health in humans.

## 5. Conclusions

In summary, this study investigated the correlation between the structure of fecal microbiota and redox potential in healthy and diarrheal pigs. The results revealed significant associations between fecal redox potential and the abundance of *Limosilactobacillus*, *Escherichia-Shigella*, and *Prevotella*, which indicates their active roles in shaping the chemical environment in the gut. Moreover, an increase in redox potential following diarrhea was accompanied by a rise in *Escherichia-Shigella* abundance, which suggests that regulating intestinal redox potential may effectively reduce pathogenic bacteria and promote intestinal health.

## Figures and Tables

**Figure 1 antioxidants-13-00096-f001:**
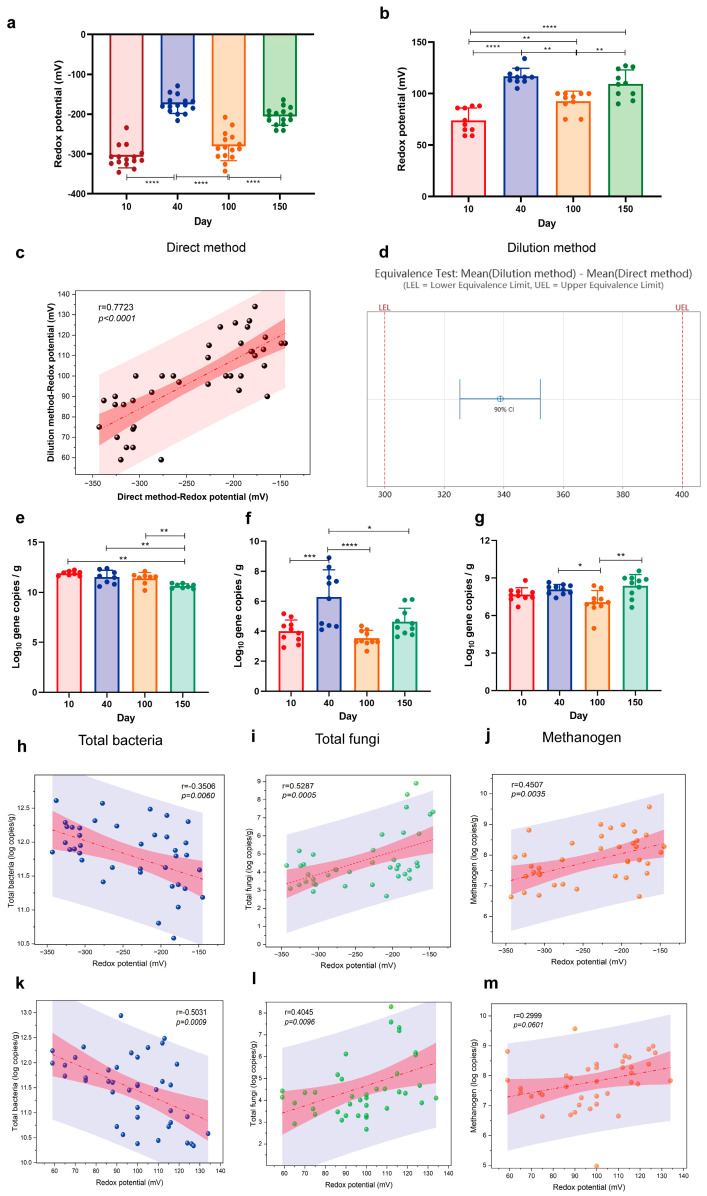
The correlation of fecal redox potential with the numbers of bacteria, methanogens, and fungi in healthy pigs. The changes in redox potential in pigs at different growth stages using direct (**a**) and dilution (**b**) methods. (**c**) Correlation analysis of redox potential measured using direct and dilution methods. (**d**) The equivalence test of direct and dilution methods. (**e**–**g**) Real-time quantitative pCR for total bacteria, fungi, and methanogens. (**h**–**j**) Correlation analysis of the numbers of total bacteria, fungi, and methanogens with redox potential measured using direct method. (**k**–**m**) Correlation analysis of the numbers of total bacteria, fungi, and methanogens with redox potential measured using dilution method. (* *p* < 0.05; ** *p* < 0.01; *** *p* < 0.001; **** *p* < 0.0001; the pink areas represent 95% confidence intervals and the blue areas represent 95% prediction intervals).

**Figure 2 antioxidants-13-00096-f002:**
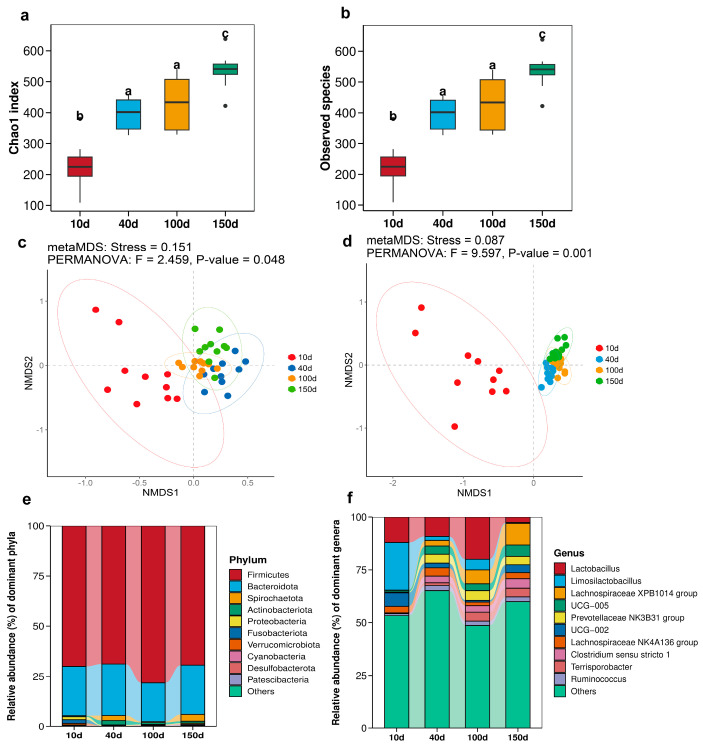
Bacterial communities in feces of pigs at different growth stages. The Chao1 index (**a**) and observed species (**b**) of pig feces on different days of age. Non-metric multidimensional scaling was conducted based on pig fecal microbiota at the phylum (**c**) and genus (**d**) levels. The dots of different hues represent identical days of age, and the circles of the same color represent 95% confidence intervals. The microbial composition at the phylum (**e**) and genus (**f**) levels. Different letters on the column indicate significant differences, while the same letters indicate non-significant differences.

**Figure 3 antioxidants-13-00096-f003:**
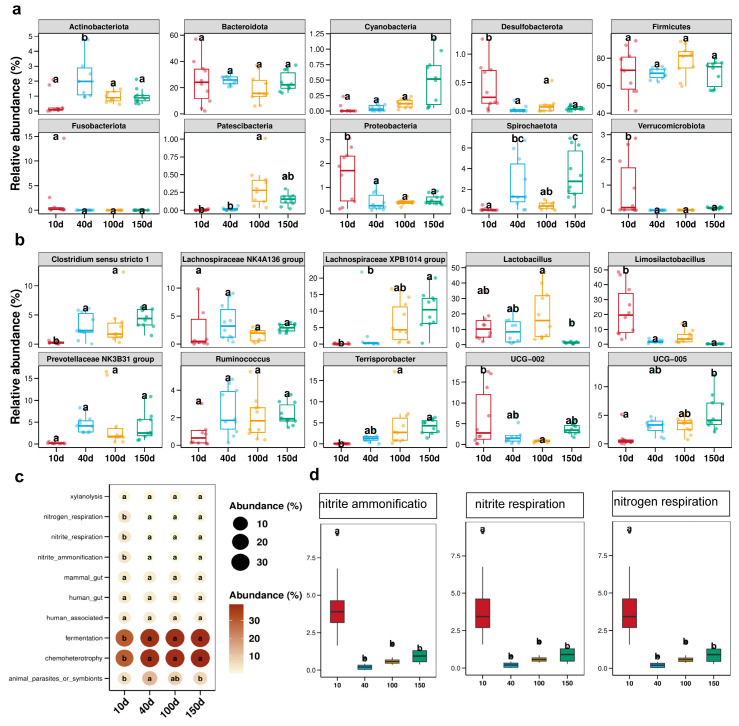
The phylum (**a**) and genus (**b**) levels were utilized for conducting statistical analysis on the top 10 microbiota present in pig feces across different growth stages. (**c**) Functional prediction analysis of pig feces was conducted utilizing FAPROTAX. (**d**) Nitrite ammonification, nitrite respiration, and nitrogen respiration by intestinal microbiota. Different letters on the column indicate significant differences, while the same letters indicate non-significant differences.

**Figure 4 antioxidants-13-00096-f004:**
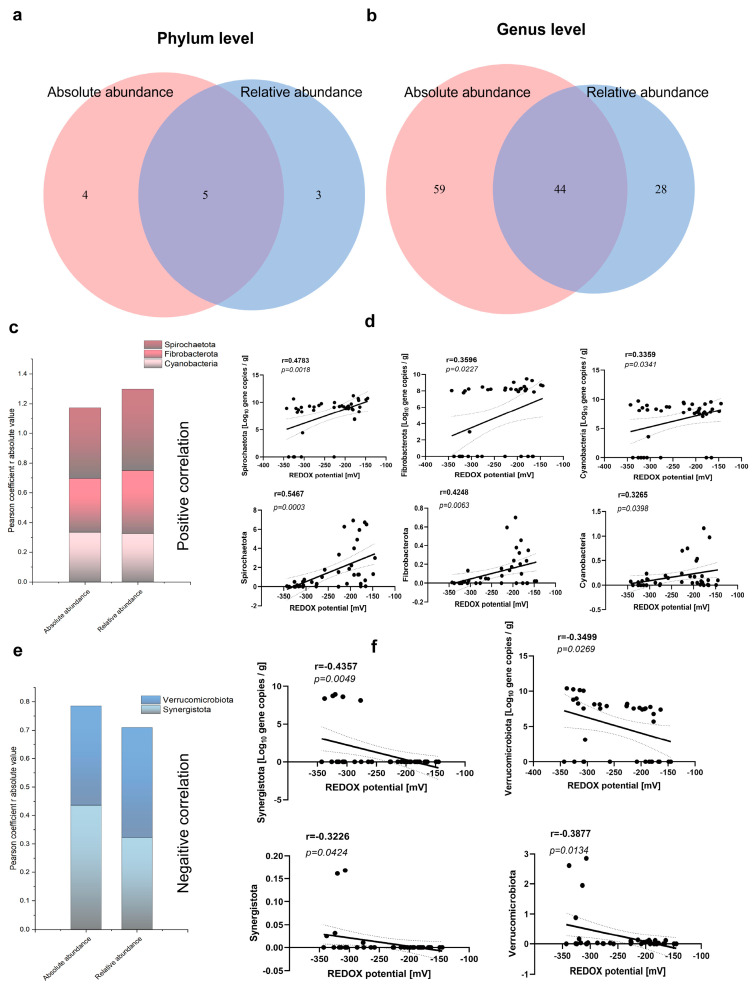
Correlation of fecal redox potential with the abundance of bacterial communities at the phylum and genus levels. The Venn diagram revealed the absolute and relative abundances in correlation with redox potential at the phylum (**a**) and genus (**b**) levels. The common bacteria have a positive (**c**,**d**) and negative (**e**,**f**) correlation with redox potential at the phylum level. The dotted areas represent 95% confidence intervals.

**Figure 5 antioxidants-13-00096-f005:**
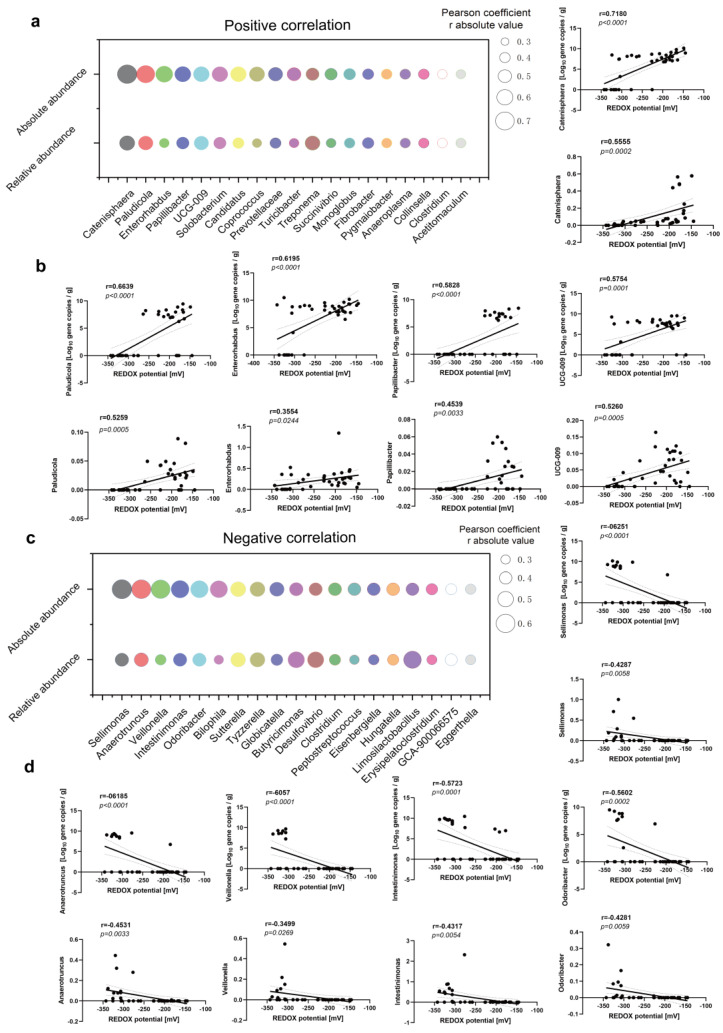
The common bacteria have a positive (**a**,**b**) and negative (**c**,**d**) correlation with redox potential at the genus level. The dotted areas represent 95% confidence intervals.

**Figure 6 antioxidants-13-00096-f006:**
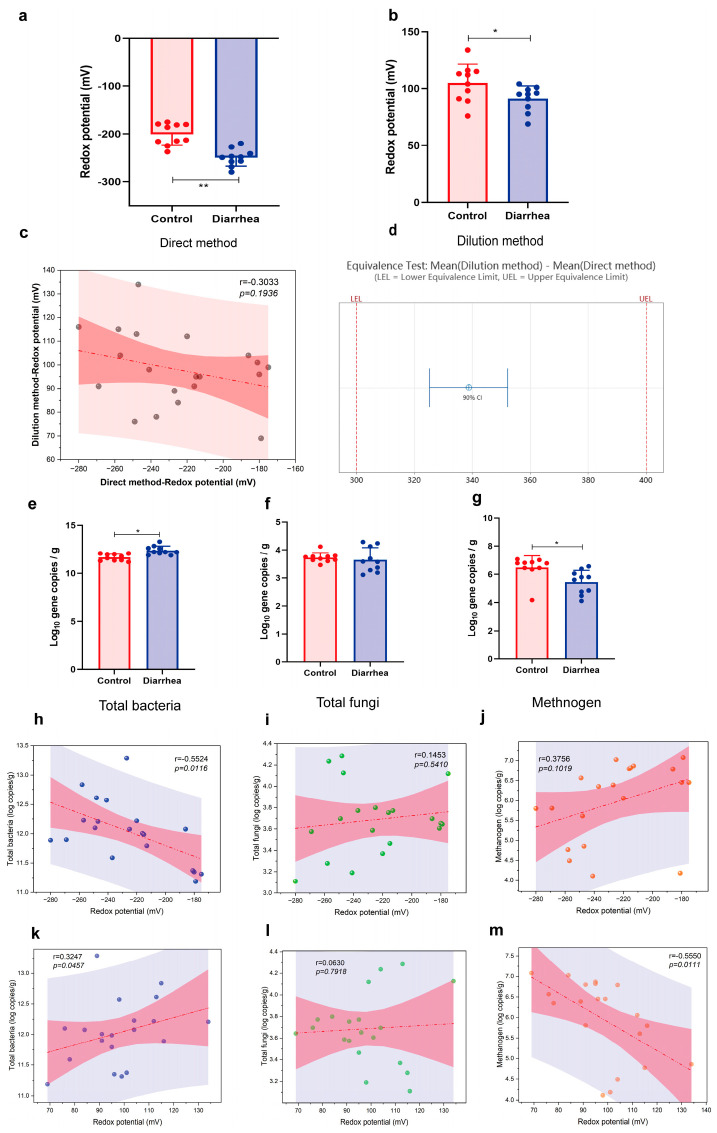
Fecal redox potential and the number of bacteria, methanogens, and fungi in diarrheal piglets. The redox potential of control and diarrhea groups was measured using the direct method (**a**) and the dilution method (**b**). * *p* < 0.05; ** *p* < 0.01. (**c**) Correlation analysis of deferent redox potential measured using direct and dilution methods. (**d**) The equivalence test of direct and dilution methods. (**e**–**g**) Real-time quantitative PCR for total bacteria, fungi, and methanogens. (**h**–**j**) Correlation analysis of the total bacteria, fungi, and methanogens with redox potential was conducted using the direct method. (**k**–**m**) Correlation analysis of the total bacteria, fungi, and methanogens with redox potential was conducted using the dilution method.

**Figure 7 antioxidants-13-00096-f007:**
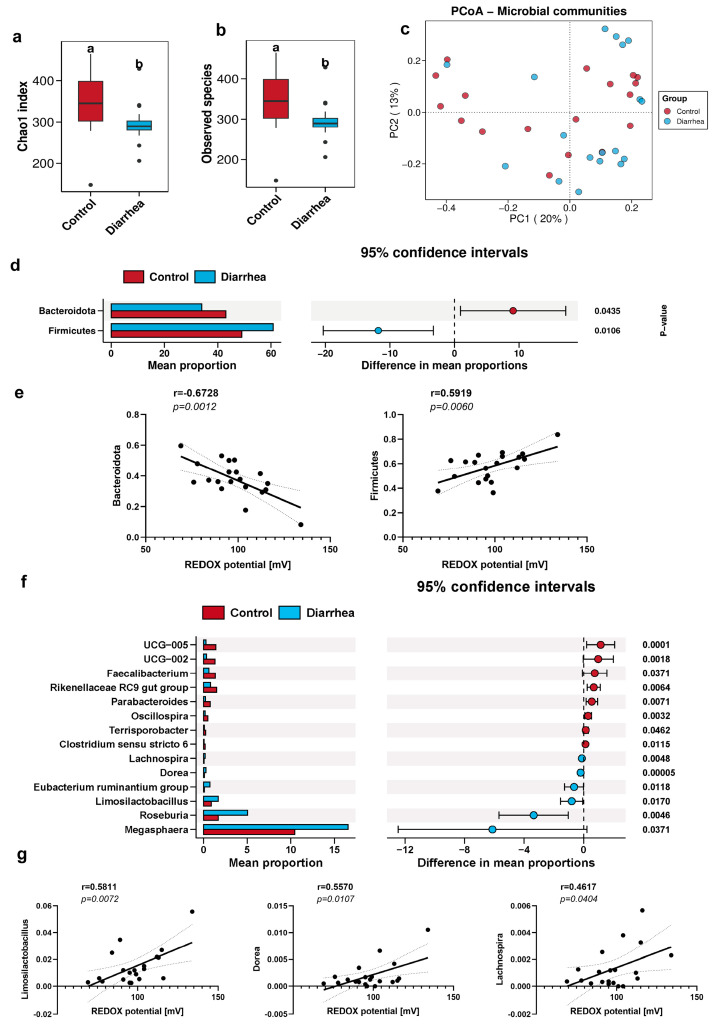
The Chao1 coefficient (**a**) and observed species (**b**) of pig feces in diarrheal and healthy piglets. (**c**) Principal coordinates analysis plot based on the pig feces in diarrheal and healthy piglets. The differential microbiota at the phylum (**d**) and genus (**f**) levels was analyzed using the Wilcoxon test. Correlation analysis between the relative abundance of different bacteria and redox potential at the phylum (**e**) and genus (**g**) levels. The dotted areas represent 95% confidence intervals.

**Figure 8 antioxidants-13-00096-f008:**
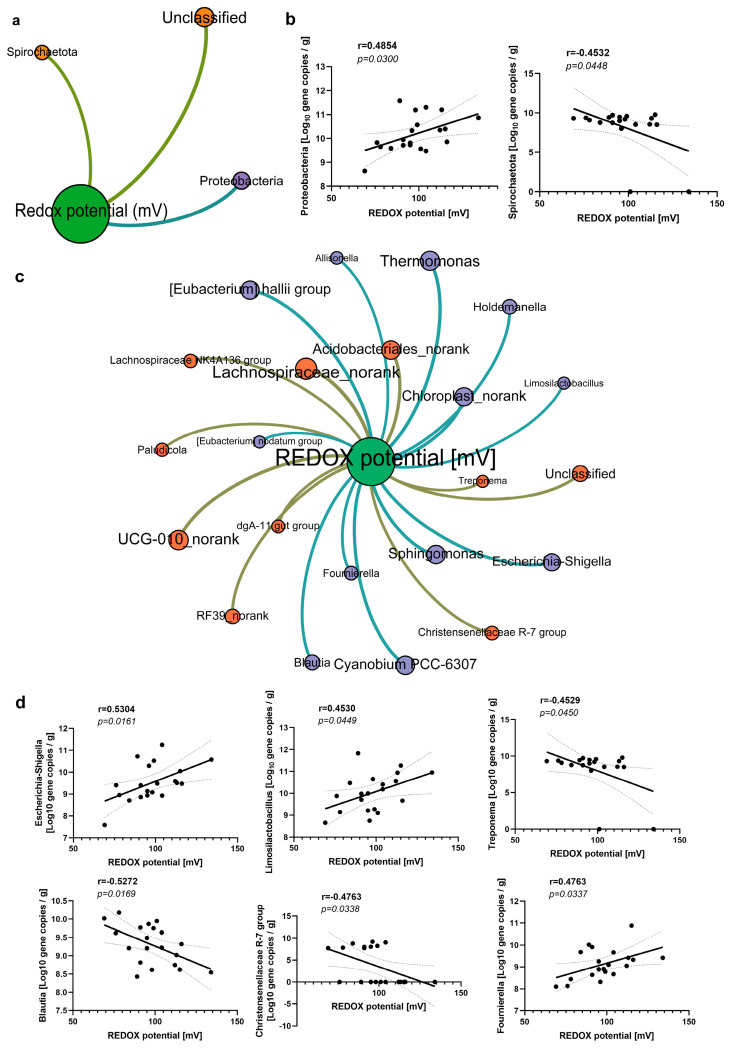
Correlation analysis between bacteria of absolute abundance and redox potential at the phylum (**a**,**b**) and genus (**c**,**d**) levels. The green node represents the redox potential, purple nodes represent the bacteria that are negatively correlated with the redox potential, orange nodes represent the bacteria that are positively correlated with the redox potential, and the size of the nodes indicates the strength of the correlation. The dotted areas represent 95% confidence intervals.

**Figure 9 antioxidants-13-00096-f009:**
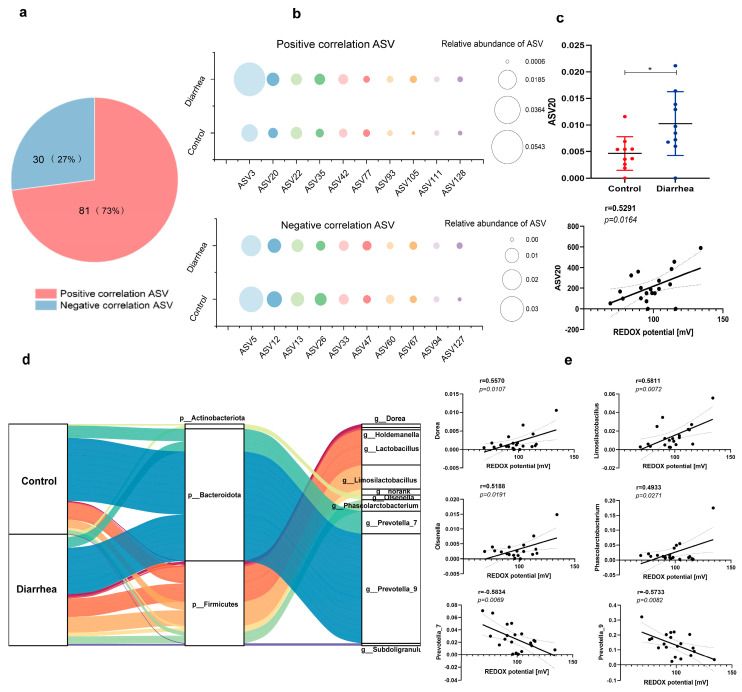
The association between redox potential and the fecal microbial community of pigs with diarrhea at the ASV level. (**a**) The pie chart shows ASVs in relation to redox potential. (**b**) The bubble plots show ASVs that are positively and negatively correlated with the redox potential. (**c**) Relative abundance of ASV20 in diarrhea group and control group and its correlation with redox potential. * *p* < 0.05. (**d**) The annotated mulberry plot of ASVs related to redox potential. (**e**) Correlation analysis between genera annotated by ASVs and redox potential. The dotted areas represent 95% confidence intervals.

## Data Availability

Data are contained within the article.

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
