# Peer review of "The Fecal Redox Potential in Healthy and Diarrheal Pigs and Their Correlation with Microbiota"

_antioxidants, 2024, doi:10.3390/antiox13010096_

Round 1

Reviewer 1 Report

Comments and Suggestions for Authors

 Dear Authors

I present my comments below. There are quite a few of them, but the results are very interesting. I would suggest reediting the work because it is worth publishing.

1.      In subsection 2.1 I lack information on how and from where the stool samples were taken – e.g. whether from the distal section of the colon or from the litter;

2.      The discussion was not substantive and general – they were too sparing in their own results (correlations);

3.      The arguments or thoughts presented refer to physiological states, not to pathology such as diarrhea – after all, the moment of diarrhea occurs, as a rule, there is a therapeutic effect (antibiotics or sulfonamides);

4.      And what about antibiotic growth promoters;

5.      We are talking about strains of probiotic, exopolysaccharide type microorganisms, and where there is room for strains of saprophytic bacteria not always beneficial for the macroorganism;

6.      A lot of generalizations to nutrients in the digestive content, and what about undesirable substances – have they been eliminated (as it has been checked);

7.      And what about the succolian states of the intestines of certain conditions;

8.      These may be empirical studies, but they are not very scientific, using many generalities that are not supported by scientific evidence. When extrapolating on the subject, one should use various substantive evidence (not only physiological, but also pathological) and support oneself with one's own results. For this reason, it would be worthwhile to reduce the number of generalities and focus on a few and discuss them in detail.

Author Response

Dear Reviewer,

We appreciate your careful reading and positive response to our manuscript. We have carefully considered the comments and have revised the manuscript accordingly. Our itemized response list as follows:

  1. In subsection 2.1 I lack information on how and from where the stool samples were taken– g. whether from the distal section of the colon or from the litter;

Response: Thank you for your thoughtful advice. All samples were taken from the litter and we provide additional explanations in line 95 of Materials and methods.

  1. The discussion was not substantive and general – they were too sparing in their own results (correlations);

Response: The authors would like to thank the reviewer for this suggestion. We have refined and modified the discussion.

  1. The arguments or thoughts presented refer to physiological states, not to pathology such as diarrhea – after all, the moment of diarrhea occurs, as a rule, there is a therapeutic effect (antibiotics or sulfonamides);

Response: The authors would like to thank the reviewer for the comment on the problem. Piglets exhibiting spontaneous post-weaning diarrhea and unaffected counterparts were carefully chosen from distinct litters within a single swine facility, ensuring that no antibiotic treatment was given before or after the manifestation of diarrheal symptoms.

  1. And what about antibiotic growth promoters;

Response: Thank you very much for your careful comments. None of the piglets received antibiotics or probiotic products throughout the study period.

  1. We are talking about strains of probiotic, exopolysaccharide type microorganisms, and where there is room for strains of saprophytic bacteria not always beneficial for the macroorganism;

Response: Thank you for your thoughtful advice. We have expanded and corrected the discussion in this section

  1. A lot of generalizations to nutrients in the digestive content, and what about undesirable substances– have they been eliminated (as it has been checked);

Response: Thank you very much for your careful comments. The undesirable substances have been eliminated.

  1. And what about the succolian states of the intestines of certain conditions;

Response: Thank you very much for your thoughtful comments. The question mentioned is exactly the idea of our next research. This paper only focuses on the relationship between intestinal flora and redox potential in different states.

  1. These may be empirical studies, but they are not very scientific, using many generalities that are not supported by scientific evidence. When extrapolating on the subject, one should use various substantive evidence (not only physiological, but also pathological) and support oneself with one's own results. For this reason, it would be worthwhile to reduce the number of generalities and focus on a few and discuss them in detail.

Response: Thank you very much for the opinion. Although studies have primarily focused on the succession patterns of animal intestinal flora, limited attention has been given to understanding the succession patterns of potential flora at different growth stages. Therefore, this study aims to investigate the variations in redox potential during different growth stages and episodes of diarrhea, while also analyzing its correlation with intestinal flora. These findings will serve as a foundation for future research on the contribution of specific flora to redox potential and targeted regulation of intestinal health. This is also the shortcoming of our research, which will be further explored in our subsequent research.

Reviewer 2 Report

Comments and Suggestions for Authors

This work is about the importance of fecal redox potential in healthy and diarrheal pigs and the correlation with their microbiota. Authors claim that results suggest that direct and dilution methods are suitable for detecting gut redox potential in healthy pigs, while the dilution method is more suitable for diarrheal pigs. The findings on correlation of microbiota  with redox potential offers novel insights for targeted modulation of intestinal health.

The experiments are well organized and the results seem to support the conclusion. Overall, it is a significant study on this matter. However, in order to be publish, in my opinion there are some major and minor changes to be addressed:

Major changes:

1)      Page 2, Line 47: The sentence “Currently, the measurement of intestinal redox potential can be categorized into two groups. One group involves directly measuring the intestinal redox potential during anaerobic operations (Direct method) [6].” should be reformulated. These are not groups, these are methodologies. The authors speak about two groups and then only speak about one. More than that (like previously mentioned these are not groups but methods. Please reformulate the para graph.

2)      Page 2, Line 75: Reformulate the sentence “When diarrhea occurs in the body, there is a significant increase in the intestinal redox potential”. The diarrhea occurs in the body? Please use a more scientific language.

3)      Page 2, Line 85: Please clarify the sentence “The study will offer novel ideas and methodologies for regulating gut microbiota and health of animals.”. It is not clear how the study offer methodologies for regulating gut microbiota. It clearly offers understanding not solutions.

4)      Page 2, Materials and Methods, Animals, experimental design and sampling: Were these studies ethically approved? Which ethics committee approved it? How was diarrhea induced? What were animal facilities conditions for that?

5)      Page 2, Materials and Methods, Redox potential measurement: These are direct and dilution methods not Method 1 and 2. Please reformulate this section without using the first person (our). Also 7 reference must be in the end of the phrase after “eletrodes”.

6)      Page 3, Line 108: Please reformulate the section “DNA extraction and real-time quantitative PCR”. Language is not appropriate (e.g. in conjunction) and also references are not correctly indicated (e.g. Jing 2022 14) once the authors are no coherent thought the text in the way they refer other works.

7)      Pag 3. Line 125: Explain what do you mean by “passing quality inspection”.

8)      Pag 3, Statistical analysis: How many samples? Just once? More than once for animal? In the same day or not? Please explain details.

9)      Pag 4 and throughout the entire manuscript: Do not use method 1 and 2. Use direct and dilution method.

10)   Page 6, Figure 2: It is very difficult to see and analyse. Try to reorganize the figures to make results easier to read. Eventually divide it.

11)   Figure 3, 5 and 5: Impossible to read. Try to reorganize and eventually divide.

12)   The discussion seems good however as sometimes is very difficult to see the figures it is a little bit difficult to say.

Minor changes:

1)   Page 1, Line 12:  Where it is “The study measured” it should be “This study measured”.

2)   Page 1 Line 38: The sentence “The gut environment is characterized by a complex mixture of bacteria, chyme and various metabolites, which together resemble black holes.” Should be changed once the underlined part is scientifically correct.

3) Page 2, Line 62: The “Gryaznova [8] et al.'s” must be corrected. The 8 must be in the end not in the middle. Also confirm the et al.’s use.

Page 5, Fig1: Caption is not readable.

Page 6, Fig 2 and 5: Caption is unformatted.

This is an important study that after these changes can be considered for publication.

Author Response

Dear Reviewer,

We appreciate your careful reading and valuable comments to our manuscript. We have carefully considered the comments and have revised the manuscript accordingly. Our itemized response list as follows:

Major changes:

Page 2, Line 47: The sentence “Currently, the measurement of intestinal redox potential can be categorized into two groups. One group involves directly measuring the intestinal redox potential during anaerobic operations (Direct method) [6].” should be reformulated. These are not groups, these are methodologies. The authors speak about two groups and then only speak about one. More than that (like previously mentioned these are not groups but methods. Please reformulate the para graph.

Response: The authors would like to thank the reviewer for this suggestion. We have reformulated the paragraph.

Page 2, Line 75: Reformulate the sentence “When diarrhea occurs in the body, there is a significant increase in the intestinal redox potential”. The diarrhea occurs in the body? Please use a more scientific language.

Response: The comments of the reviewers are apt. We have deleted “in the body” to make the language more scientific.

Page 2, Line 85: Please clarify the sentence “The study will offer novel ideas and methodologies for regulating gut microbiota and health of animals.”. It is not clear how the study offer methodologies for regulating gut microbiota. It clearly offers understanding not solutions.

Response: The authors would like to thank the reviewer for this suggestion. The point of this article is just as the reviewer said. We have changed “methodlogies” to “understanding”.

Page 2, Materials and Methods, Animals, experimental design and sampling: Were these studies ethically approved? Which ethics committee approved it? How was diarrhea induced? What were animal facilities conditions for that?

Response: The comments of the reviewers are apt. These studies were ethically approved. The animal study protocol was approved by the Animal Care and Use Committee of Nanjing Agricultural University (SYXK2019-0066). Piglets exhibiting spontaneous post-weaning diarrhea and unaffected counterparts were carefully chosen from distinct litters within a single swine facility. We have added details in 2.1 Animals, experimental design and sampling to state the specific situation.

Page 2, Materials and Methods, Redox potential measurement: These are direct and dilution methods not Method 1 and 2. Please reformulate this section without using the first person (our). Also 7 reference must be in the end of the phrase after “eletrodes”.

Response: Thank you for your thoughtful advice and we have revised it.

Page 3, Line 108: Please reformulate the section “DNA extraction and real-time quantitative PCR”. Language is not appropriate (e.g. in conjunction) and also references are not correctly indicated (e.g. Jing 2022 14) once the authors are no coherent thought the text in the way they refer other works.

Response: The authors would like to thank the reviewer for this suggestion. We have corrected the misrepresentation and the reference.

Pag 3. Line 125: Explain what do you mean by “passing quality inspection”.

Response: Thank you for your thoughtful advice. “passing quality inspection”means we detected the correct size and concentration of extracted DNA by PCR. We have reformulated the paragraph.

Pag 3, Statistical analysis: How many samples? Just once? More than once for animal? In the same day or not? Please explain details.

Response: Thank you for your thoughtful advice. Each group had 10 samples. We collected fresh fecal samples from different groups during the same period and at the same time we described them in detail in the materials and methods.

Pag 4 and throughout the entire manuscript: Do not use method 1 and 2. Use direct and dilution method.

Response: The authors would like to thank the reviewer for the suggestion. We have checked and revised the full text.

Page 6, Figure 2: It is very difficult to see and analyse. Try to reorganize the figures to make results easier to read. Eventually divide it.

Response: Thank you for your thoughtful advice. We have reorganized the figures and devided Figure 2 into two figures.

Figure 3, 5 and 5: Impossible to read. Try to reorganize and eventually divide.

Response: The authors would like to thank the reviewer for this suggestion. We have reorganized the figures and devided Figure 3 and 5 into two figures.

The discussion seems good however as sometimes is very difficult to see the figures it is a little bit difficult to say.

Response: Thank you very much for your careful comments. We have revised the discussion to include figures citations.

Minor changes:

Page 1, Line 12: Where it is “The study measured” it should be “This study measured”.

Response: Yes and thank you for your careful advice. We have changed “The study measured”to “This study measured”.

Page 1 Line 38: The sentence “The gut environment is characterized by a complex mixture of bacteria, chyme and various metabolites, which together resemble black holes.” Should be changed once the underlined part is scientifically correct.

Response: Thank you for your thoughtful advice. We delete this unscientific statement.

Page 2, Line 62: The “Gryaznova [8] et al.'s” must be corrected. The 8 must be in the end not in the middle. Also confirm the et al.’s use.

Response: Yes and thank you for your careful advice. We have revised this.

Page 5, Fig1: Caption is not readable.

Response: The authors would like to thank the reviewer for this suggestion. We adjusted the caption based on the figures and hoped it would become readable

Page 6, Fig 2 and 5: Caption is unformatted.

Response: Thank you for your thoughtful advice. We have revised this.

This is an important study that after these changes can be considered for publication.

Reviewer 3 Report

Comments and Suggestions for Authors

General comments

The subject is original and on the whole clearly presented. However, the following points need to be clarified:

1. add the composition of the feed used

2. better specify the conditions under which the animals were reared: individual rearing, group rearing, type of soil, etc., the existence of previous antiparasitic or anti-infectious treatments, etc. Growth curves and quantities of feed ingested should be provided as additional files,

3. better define how the "diarrhoea" group was formed. Has specific aetiological research been carried out into the cause of this diarrhoea? Did this pathology affect the subsequent growth of the animals?

4. give the calibration method for the equipment used to measure redox potential

5. Specify whether the microorganisms are expressed in g of dry matter.

Other points that could be improved :

6.            The abstract should provide more details to encourage reading of the manuscript.

7.            An equivalence test of the analytical methods used could be carried out in addition to the Pearson coefficient.

8.            The discussion should address the causes of diarrhoea.

Detailed comments

L12: give the age or weight of the animals, the type of feed used and the number of animals in each group.

L14: state which methods were used. Specify that the study concerns faecal microbiota and not gut microbiota, which varies according to the portion of the digestive tract analysed.

L89: Specify the conditions under which the animals were reared, the existence of previous treatments, etc. It is essential to better define the "diarrhoea" group. What criteria are used to assign an animal to this group: % moisture content of excrement, clinical signs? Are there any exclusion criteria?

L96-105: Specify how the equipment was calibrated. Were measurements repeated after a certain time interval, for example?

L126: Are the results expressed in g of dry matter?

L155-175: the existence of a significant correlation does not mean that the two methods are equivalent. A two one-sided test could be carried out.

Figure 2 : « ammonification » is correct

Figure 4 : « Methanogen » is correct

Author Response

Dear Reviewer,

We appreciate your careful reading and valuable comments to our manuscript. We have carefully considered the comments and have revised the manuscript accordingly. Our itemized response list as follows:

Comments

  1. add the composition of the feed used

Response: Thank you for your thoughtful advice. We have added the composition of the feed used in Supplementary material.

  1. better specify the conditions under which the animals were reared: individual rearing, group rearing, type of soil, etc., the existence of previous antiparasitic or anti-infectious treatments, etc. Growth curves and quantities of feed ingested should be provided as additional files,

Response: The comments of the reviewers are apt. This part of the content is explained in detail in the Materials and methods (specify the conditions under which the animals were reared).

  1. better define how the "diarrhoea" group was formed. Has specific aetiological research been carried out into the cause of this diarrhoea? Did this pathology affect the subsequent growth of the animals?

Response: Thank you for your thoughtful advice. Diarrhea occurs naturally after weaning. We collected fecal samples from pigs of different ages at the same farm during the same period, which are described in detail in the materials and methods.

  1. give the calibration method for the equipment used to measure redox potential

Response: The authors would like to thank the reviewer for the suggestion. The OPR engineer of the ST300 we used explained that this electrode does not need to be calibrated because of its built-in standard electrode. After each sample is measured, the electrode is repeatedly rinsed with deionized water, and the next sample is measured after a light wipe with a paper towel.

  1. Specify whether the microorganisms are expressed in g of dry matter.

Response: The authors would like to thank the reviewer for this suggestion. We have specified results are expressed as the numbers of 16S rRNA or ITS gene copies per gram of wet feces in 2.3 DNA extraction and real-time quantitative PCR.

Other points that could be improved :

  1. The abstract should provide more details to encourage reading of the manuscript.

Response: The authors would like to thank the reviewer for this suggestion. We offered more details in the abstract.

  1. An equivalence test of the analytical methods used could be carried out in addition to the Pearson coefficient.

Response: Thank you for your thoughtful advice. We have carried out an equivaalence test of the analytical methods.

  1. The discussion should address the causes of diarrhoea.

Response: The authors would like to thank the reviewer for this suggestion. We have address the causes of diarrhoea.

Detailed comments

L12: give the age or weight of the animals, the type of feed used and the number of animals in each group.

Response: Thank you for your thoughtful advice. We have made detailed additions in the materials and methods.

L14: state which methods were used. Specify that the study concerns faecal microbiota and not gut microbiota, which varies according to the portion of the digestive tract analysed.

Response: Thank you for your thoughtful advice. Yes we have specified that the study concerns faecal microbiota and not gut microbiota in the abstract and Materials and methods.

L89: Specify the conditions under which the animals were reared, the existence of previous treatments, etc. It is essential to better define the "diarrhoea" group. What criteria are used to assign an animal to this group: % moisture content of excrement, clinical signs? Are there any exclusion criteria?

Response: The authors would like to thank the reviewer for this suggestion. We have added this material in Materials and Methods. the animals had not received any medication or probiotic products before and after delivery. The "diarrhoea" group: Clinical symptoms of diarrhea, stool water content of more than 50%.

L96-105: Specify how the equipment was calibrated. Were measurements repeated after a certain time interval, for example?

Response: Yes and thank you for your careful advice. The OPR engineer of the ST300 we used explained that this electrode does not need to be calibrated because of its built-in standard electrode. After each sample is measured, the electrode is repeatedly rinsed with deionized water, and the next sample is measured after a light wipe with a paper towel.

L126: Are the results expressed in g of dry matter?

Response: Thank you for your thoughtful advice. We have specified results are expressed as the numbers of 16S rRNA or ITS gene copies per gram of wet feces in 2.3 DNA extraction and real-time quantitative PCR.

L155-175: the existence of a significant correlation does not mean that the two methods are equivalent. A two one-sided test could be carried out.

Response: The comments of the reviewers are apt. We have carried out an equivaalence test of the analytical methods.

Figure 2 : « ammonification » is correct

Response: The authors would like to thank the reviewer for this suggestion. We have revised this.

Figure 4 : « Methanogen » is correct

Response: The authors would like to thank the reviewer for this suggestion. We have revised this.

Round 2

Reviewer 2 Report

Comments and Suggestions for Authors

This work is about the importance of fecal redox potential in healthy and diarrheal pigs and the correlation with their microbiota. Authors claim that results suggest that direct and dilution methods are suitable for detecting gut redox potential in healthy pigs, while the dilution method is more suitable for diarrheal pigs. The findings on correlation of microbiota with redox potential offers novel insights for targeted modulation of intestinal health.

The authors have answered reviewers comments and made the requested changes. However, small adjustments are needed; these are formatting issues rather than issues with the text itself.

The manuscript is publishable after the following changes in formatting:

Figure 2 caption

Figure 3 caption

Check Figure 5 quality to make sure it is readable

Figure 7 caption

Author Response

Dear Reviewer:

We very much appreciate your careful reading and positive response to our manuscript. We have carefully considered the comments and have revised the manuscript accordingly. Our itemized response list as follows:

Comments

Figure 2 caption

Response: The authors would like to thank the reviewer for this suggestion. Our caption of Figure 2 has been revised and supplemented.

Figure 3 caption

Response: The authors would like to thank the reviewer for this suggestion. Our caption of Figure 3 has been revised and supplemented.

Check Figure 5 quality to make sure it is readable

Response: The authors would like to thank the reviewer for this suggestion. We increased the resolution of picture 5 to 1000dpi and increased the length and width of the picture to make sure it is readable.

Figure 7 caption

Response: The authors would like to thank the reviewer for this suggestion. Our caption of Figure 7 has been revised and supplemented.

Reviewer 3 Report

Comments and Suggestions for Authors

Thank you for the clarifications made to the manuscript

Author Response

Thank you for the clarifications made to the manuscript.

Response: We appreciate your careful reading and positive response to our manuscript.